# Photocatalysis and Li-Ion Battery Applications of {001} Faceted Anatase TiO₂-Based Composites

**Anuja Bokare \* and Folarin Erogbogbo \***

Department of Biomedical Engineering, San José State University, 1 Washington Square, San José, CA 95112, USA
\* Correspondence: anujabokare@gmail.com (A.B.); folarin.erogbogbo@sjsu.edu (F.E.)

**Abstract:** Anatase TiO₂ are the most widely used photocatalysts because of their unique electronic, optical and catalytic properties. Surface chemistry plays a very important role in the various applications of anatase TiO₂ especially in the catalysis, photocatalysis, energy conversion and energy storage. Control of the surface structure by crystal facet engineering has become an important strategy for tuning and optimizing the physicochemical properties of TiO₂. For anatase TiO₂, the {001} crystal facets are the most reactive because they exhibit unique surface characteristics such as visible light responsiveness, dissociative adsorption, efficient charge separation capabilities and photocatalytic selectivity. In this review, a concise survey of the literature in the field of {001} dominated anatase TiO₂ crystals and their composites is presented. To begin, the existing strategies for the synthesis of {001} dominated anatase TiO₂ and their composites are discussed. These synthesis strategies include both fluorine-mediated and fluorine-free synthesis routes. Then, a detailed account of the effect of {001} facets on the physicochemical properties of TiO₂ and their composites are reviewed, with a particular focus on photocatalysis and Li-ion batteries applications. Finally, an outlook is given on future strategies discussing the remaining challenges for the development of {001} dominated TiO₂ nanomaterials and their potential applications.

**Keywords:** high energy TiO₂ facet; {001} TiO₂ facet; TiO₂-graphene composite; doping; photocatalysis; Li-ion battery anode





## 1. Introduction

The rising demand for energy and the increase in environmental pollution have become extremely serious issues in recent years [1,2]. Harnessing the sun's energy to produce electricity and to remediate environmental pollution through the use of advanced nanomaterials has proven to be a promising solution to the world's energy crisis [3,4]. One of the first technologies that come to mind when discussing solar energy is photocatalysis. Photocatalysis relies on using the sunlight to promote the degradation of organic pollutants [5,6]. Among a wide spectrum of semiconductors, TiO2 is the most efficient photocatalyst due to its chemical stability, non-toxicity, strong oxidizing power, biocompatibility, large surface area, corrosion resistivity and cost effectiveness [7–9].

The intense interest in titanium dioxide (TiO₂) as a photocatalyst has spurred the successful synthesis and extensive investigations of a variety of its crystal facets. Exploring the application of the high energetic {001} facets of titania is a recent venture in TiO₂ photocatalysis [10–13]. A natural anatase crystal typically can exhibit many crystal facets, as shown in Figure 1 [14,15]. Among them, {001} surfaces are highly desired because they are theoretically predicted to possess more active sites (five coordinated Ti⁺) and higher surface energy (0.90 J/m²) relative to other energetically more favorable facets such as {100} 0.53 J/m² or {101} 0.44 J/m² [16]. Both theoretical and experimental studies demonstrate that the photocatalytic activity of the anatase {001} facets is higher than that of the thermodynamically stable {101} facets [17–19]. Apart from photocatalytic studies, many other promising applications such as photo- or electrocatalysis, photoelectrochemical

or photovoltaic cells, lithium/sodium ion batteries, Li–S batteries and gas sensing can be significantly improved by morphological control and specifically exposed {001} facets on the surface [20–23]. Hence, more efforts are being made to synthesize $TiO_2$ nanomaterials with dominant high-energy {001} facets ($TiO_2$-001).

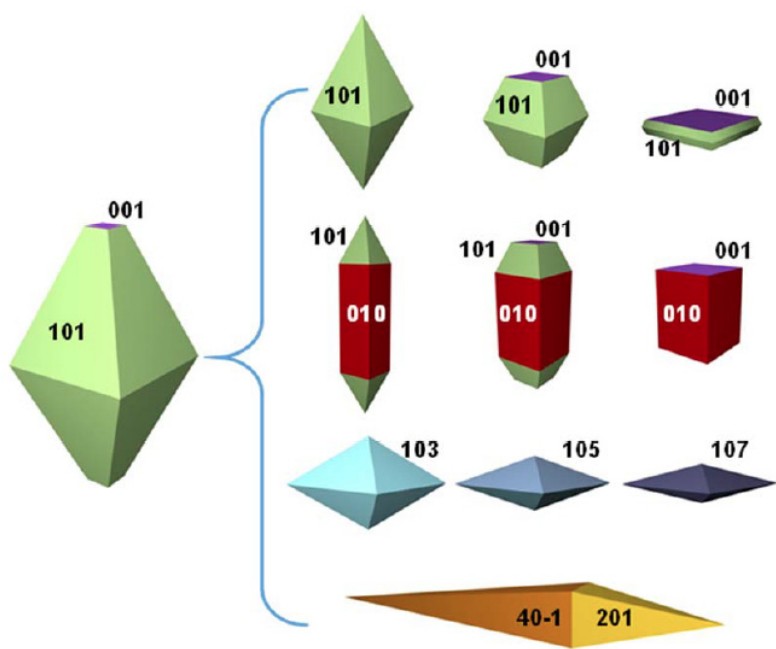

**Figure 1.** Equilibrium crystal shape of anatase $TiO_2$ through the Wulff construction and the evolved other shapes. (Reprinted with the permission of ref. [14], copyrights 2014 © American Chemical Society).

Even though the design and morphologically controlled synthesis of $TiO_2$-001 is considered to be a hot spot in scientific research, it still has some drawbacks like its wide band gap and high recombination rate [24]. These drawbacks can easily be overcome by coupling them with other materials to form $TiO_2$-001-based composites. For example, doping $TiO_2$-001 with metal (transition or rare-earth metal) or non-metal ions (C, F, S, N) can make them responsive to visible light [25]. Higher rates of recombination can be suppressed by noble metal deposition on the $TiO_2$-001 surface [26,27]. When coupled with carbon-based materials, $TiO_2$-001-based composites show excellent properties, such as high surface area, high absorptivity of dyes and high charge separation [28,29]. These properties make them applicable for many areas of science and technology ranging from adsorption, catalysis and photocatalysis to biomedicine, environmental monitoring and cleanup, energy conversion and storage, etc [30,31]. Hence, it is crucial to highlight the recent progress in the photocatalytic performance of $TiO_2$-based composites with {001} facets as a future energy material.

This review begins by explaining the role of {001} facets in improving the photocatalytic performance of $TiO_2$-001 photocatalysts. Then, a brief discussion is given on the modification of the catalysts by doping and coupling mechanisms. Further, we focus on the synthesis routes to obtain $TiO_2$-001 and modified $TiO_2$-001-based composites. This is then followed by the various applications of these composites which include environmental remediation by dye degradation, $H_2$ generation, $CO_2$ reduction and energy generation through Li-ion batteries. Finally, we present a conclusion and future scope of this emerging field which highlights the major challenges and some invigorating perspectives for future research.

## 2. Mechanism of Photoactivity Enhancement by {001} Facets in TiO₂

It has been widely recognized that the photocatalytic properties of $TiO_2$ nanomaterials can be altered by their crystal facets [32,33]. The opportunity to tune the surface properties

of anatase crystals by modifying the {001} exposed facets has been extensively reviewed in the last few years [34,35]. Anatase TiO$_2$ crystals with dominant {001} high energy facets possess characteristic surface configurations with plentiful dangling bonds and abundant surface defects [36]. Moreover, their surface chemistry can be easily modified, affecting adsorption, capacity, selectivity and surface reactivity [37]. In addition, the observed differences in the surface energy of crystallographic facets significantly affect the electronic and optical properties of the photocatalysts. Some of these properties are discussed below.

### 2.1. Properties of TiO$_2$-001 Surfaces for Photocatalysis
### 2.1.1. Dissociative Adsorption

In a recent study, it was found that the {010} facets could only absorb water molecules on its surface while the {001} facets could dissociate the water molecules, producing hydroxyl and other reactive radicals [38]. It is proposed that the {001} facets could also facilitate transfer of these charge carriers showing higher photocatalytic efficiency [39]. The dissociative adsorption of reactant molecules on {001} facets appears to reduce their activation energy and affect the reaction mechanism at the molecular level in the photocatalytic reaction. Similar chemical activity has also been observed in a wide range of organic species adsorption and other applications such as bacterial inactivation, verifying the unique surface chemistry of the {001} facet [40,41].

Many researchers have investigated the effect of dissociative adsorption on the photoactivity of TiO$_2$ crystals. They pointed out that the higher the percentage of {001} facets, the higher the photooxidation reactivity for similarly sized TiO$_2$ particles.

### 2.1.2. Charge Separation

Besides enhanced dissociative adsorption, charge separation of photogenerated charge carriers can also be mediated by {001} facets [42]. This can be mainly attributed to the unsaturated Ti atoms and defects (oxygen vacancy) present on the surface. The surface of TiO$_2$-001 nanomaterials exhibits high density of active, five-fold coordinated Ti atoms (Ti5c) and wide Ti-O-Ti bond angles, as can be seen in Figure 2a [43]. This gives rise to the generation of a higher number of oxygen vacancies on the surface. Liu et al. proposed that oxygen deficiency within TiO$_2$-001 induces the specific surface reconstruction in combination with surface chemical species, which facilitates the interfacial electron transfer and e$^-$/h$^+$ charge separation [44].

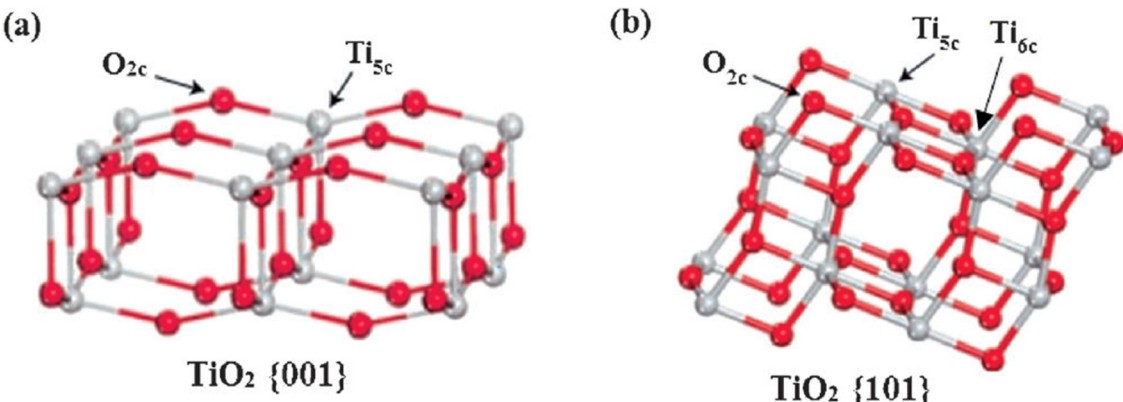

**Figure 2.** Schematic of surface atomic structure of anatase {001} and {101} crystal facets. (**a**) Large bond angles of Ti-O-Ti on the more reactive {001} surface. (**b**) Narrow bond angles of Ti-O-Ti on the less reactive {101} surface. (Reprinted with the permission of ref. [39], copyrights 2008 © Nature Publishing Group).

In addition, different atomic arrangements, surface energies and electronic structures associated with each family of TiO$_2$ facets drives selective migration of charge carriers towards different exposed facets. Generally, it is considered that the coordinately unsatu-

rated $O_{2c}$-$Ti_{5c}$-$O_{2c}$ bond on $TiO_2$-001 surface is more selective for the oxidation of adsorbed species whereas {010} facets show selectivity for the reduction reactions (Figure 3a,b) [45,46]. Hence, ideal spatial separation on these 001 (oxidation) and {010} (reduction) sites on the $TiO_2$ nanoparticle reduces the recombination of photogenerated charge carriers, thus increasing the photocatalytic efficiency. Recently, Liberto et al. [36] evaluated the role of exposed {001} and {101} crystal facets in photocatalytic processes. They detected charge-trapping centers for $Ti^{3+}$ and $O^-$ ions by performing electron spin resonance spectroscopy and concluded that the concentration of trapped holes ($O^-$ centers) increased with increasing amounts of {101} facets, whereas the amount of $Ti^{3+}$ centers for trapped electrons increased with {001} facets (Figure 3e), thus confirming the efficient charge separation on $TiO_2$-001 surfaces [47,48].

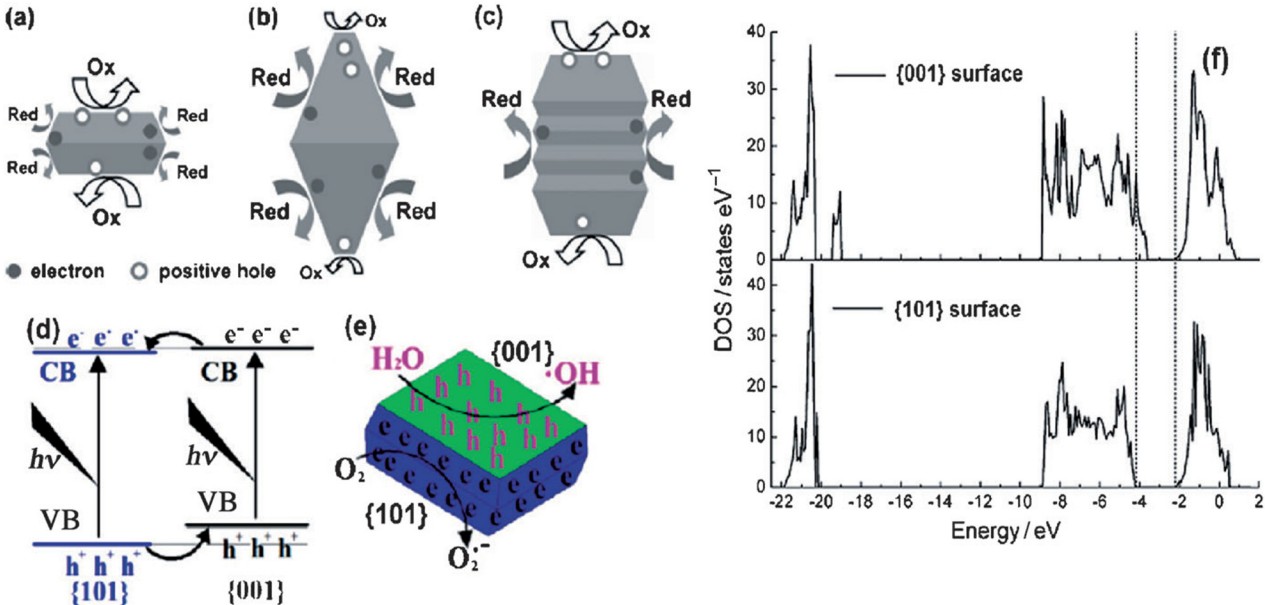

**Figure 3.** Schematic illustrations of spatial separation of redox sites on the anatase with exposed {001} and {101} crystal facets. (**a**–**c**) Different shapes of $TiO_2$ particle with a decahedral particle with {001} facets as oxidation sites and {101} facets as reducing sites. (Reprinted with the permission of ref. [49] copyrights 2009 © American Chemical Society). (**d**) Electronic band structures of the {001}{101} facets. (Reprinted with the permission of ref [50] copyrights 2013 © Elsevier). (**e**) Distribution of electrons and holes on the {001} and {101} facets. (**f**) DOS of $TiO_2$ {001} and {101} surfaces. (Reprinted with the permission of ref. [51], copyrights 2011 © John Wiley and Sons).

### 2.1.3. Optical Properties

Apart from preferential oxidation–reduction ability, the band gap of the $TiO_2$ nanomaterials would also change according to the arrangements of crystal facets on the surface [52]. Based on X-ray photoelectron spectroscopy (XPS) VB spectra and DFT electronic structure calculations, Liu et al. revealed two significant points: (1) the band gap of {001} facets was smaller than that of {101} facets, and (2) the VB maximum of {001} facets was identical to that of {101} facets and thus the CB minimum of {001} facets was lower than that of {101} facets. (Figure 3d) [53]. Hence, the UV-Visible spectrum of anatase $TiO_2$ with 72% {101} facets show blue shifts compared to that with 72% of {001} facets. This electronic band difference would certainly affect the photocatalytic activity of the catalysts in the visible light region [54]. However, up to now, no appreciable visible activity has been observed for pure anatase $TiO_2$ exposed with {001} facets. To make visible light active, these nanomaterials are coupled with other materials.

### 2.1.4. Photocatalytic Selectivity

The selectivity of the photocatalysts towards pollutants/organics is a very important aspect for its conversion/transformation. The selectivity of the $TiO_2$ can be enhanced by increasing the percentage of {001} facets on its surface. Li et al. demonstrated that the photocatalytic selectivity for the conversion of toluene to benzaldehyde can be enhanced by increasing the specific surface area of exposed {001} facets. [55]. Similarly, the rate of photoconversion of azo dyes is found to be higher for the $TiO_2$-001 materials [56].

This selectivity can be related to the special atomic configuration and associated surface terminating Ti-F bonds which dominates the interaction between {001} facets and certain organic compounds [53]. Ohtani et al. suggested that the acid strength of hydroxyl ions on {001} facets is lower than the {101} facets which is further confirmed by the zeta potential measurement. This difference in the surface charge significantly affects the adsorption properties and photocatalytic conversion of organic pollutants [57].

### 2.2. Properties of TiO2-001 Surfaces for Li-ion Battery

Lithium-ion batteries (LIBs) have been a widely applied system to store electrical energy, for both mobile and stationary applications. $TiO_2$-based anodes are the most attractive candidates for building lithium-ion batteries due to the following characteristics: (1) high specific surface area, (2) low volume change during Li ion insertion/desertions process (<4%), which is important for good cycling stability, (3) low internal resistance, (4) low intercalation potential for Li, (5) low price, (6) environmentally friendly, and (7) safe and durable.

Among other forms of $TiO_2$, $TiO_2$ nanosheets with exposed {001} facets offer very high surface areas (~170 $m^2\ g^{-1}$) which provides more lithium insertion channels, and short paths for Li-ion diffusion with high speed, which are crucial for good rate capability (Figure 4). Accordingly, the first discharge capacity of 300 mAh $g^{-1}$ was reported at 0.5 C and a reversible capacity of 150 mAh $g^{-1}$ was achieved at current rate of 10 C for $TiO_2$-001 surfaces [58]. Moreover, anatase $TiO_2$, having different percentages of (001), demonstrated different behaviors for $Li^+$ ions insertion. Hengerer et al. showed that the lithium insertion is favored on the (001) surface as compared to 101 surfaces. This can be confirmed by a higher standard rate constant for charge transfer $10^{-8}$ vs. $2 \times 10^{-9}$ cm/s and a higher chemical diffusion coefficient for $Li^+$ insertion $4 \times 10^{-13}$ vs. $2 \times 10^{-13}$ $cm^2$/s [59].

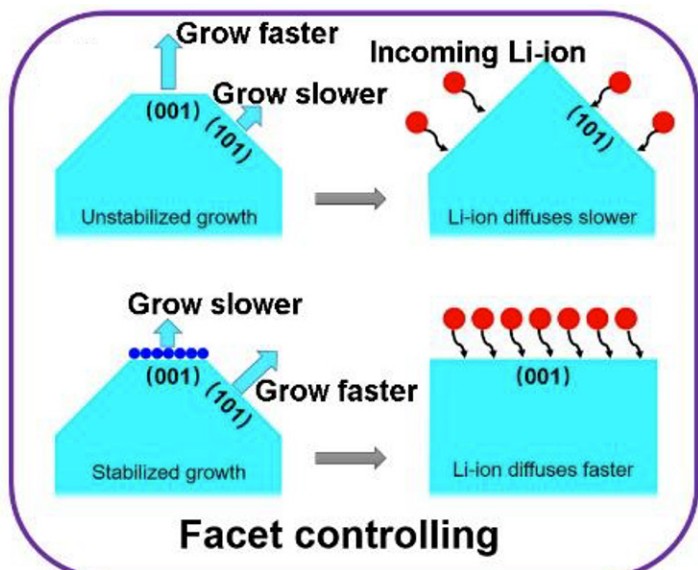

**Figure 4.** Li-ion insertion on $TiO_2$-001 surfaces [60].

Cheng et al. compared the electrochemical properties of 001 TiO$_2$ nanosheets with the Degussa P25, which can be seen in Figure 5. They revealed that the TiO$_2$-001 surfaces exhibit an excellent capacity retention at 10 C charge−discharge rate (101.9 mA h g$^{-1}$ after 100 cycles), and enhanced rate performance at 0.5−10 C current rates as compared with Degussa P25 TiO$_2$ nanoparticles (NPs) [61].

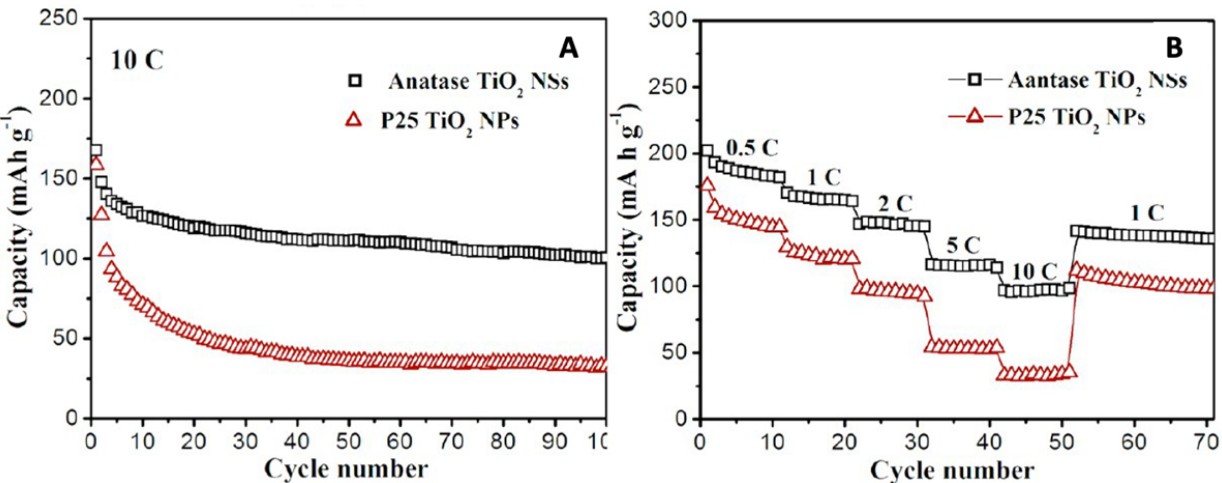

**Figure 5.** (**A**) Cycling performances of the anatase TiO$_2$ NSs and P25 TiO$_2$ NPs at 10 C. (**B**) Rate capability of the anatase TiO$_2$ NSs and P25 TiO$_2$ NPs at 0.5−10 C. (Reprinted with the permission of ref. [61], copyrights 2014 © American Chemical Society).

The specific capacity and cycling performances can be further enhanced by integrating the advantages of 001 faceted nanosheets and their highly porous organization. Yu et al. demonstrated that the TiO$_2$ hollow microspheres exhibit numerous active sites and efficient accommodation of volume changes during the charging and discharging process, leading to superior capacity and rate capability. This excellent electrochemical performance can be attributed to the hollow nature and exposed high-energy {001} facets of the 3D assembled electrode structure [20].

### 2.3. Methods to Modify the Properties of TiO$_2$-001 Surfaces

The scientists have developed many chemical methods to modify TiO$_2$ by shifting its optical response to the visible range. Some of them are given in Figure 6.

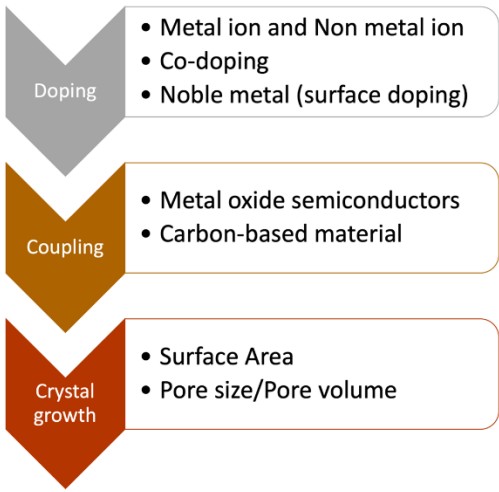

**Figure 6.** Methods to modify TiO$_2$-001 nanomaterials.

### 2.3.1. Doping

One of the most common methods to modify the properties of $TiO_2$-001 is through doping. Doping intentionally introduces impurities into pure $TiO_2$-001 semiconductors and can be done for the purpose of modulating its physical, chemical and optical properties [62]. The choice of the dopant is dependent upon its properties, such as ionic radii, conductivity and chemical stability. Here, we will discuss metal and non-metal ion doping [63].

Metal ion doping: Metal ion doping involves a substitution of Ti group by a transition metal or rare earth metal ion [64,65]. Doping with a metal ion increases the formation of $Ti^{3+}$ ions, leading to enhancement in the photocatalytic activity, as more $Ti^{3+}$ states may cause more oxygen defects which facilitate the efficient adsorption of oxygen on the titania surface. The formation of $O^{2-}$ upon chemisorptions of oxygen requires the presence of a surface defect site which can be enhanced by metal ion doping. Since the redox energy states of many metal ions lie within the band gap states of $TiO_2$, the substitution of metal ions into the $TiO_2$ introduces an intraband state close to the CB or VB edge, inducing visible light absorption at sub-band gap energies. Zhang et al. and Zhu et al. reported that Mo-doped and La-doped $TiO_2$-001 showed more enhancement in the photoactivity than the pure $TiO_2$ [66,67].

Non-metal ion doping: For non-metal doping in $TiO_2$ in anionic sites (oxygen), wide varieties of anionic species N [68], S [69], C [70], or F [71] have been used. This approach consists of the substitution of a non-metal atom for an oxygen atom in $TiO_2$. There are three different main opinions regarding modification mechanisms of $TiO_2$ doped with non-metals.

(a) Band gap narrowing: Dozzi et al. [72] found N 2p state hybrids with O 2p states in anatase $TiO_2$ doped with nitrogen because their energies are very close, and thus the band gap of N-$TiO_2$ is narrowed and able to absorb visible light.

(b) Impurity energy level: Yalda et al. [73] stated that $TiO_2$ oxygen sites substituted by nitrogen atom form isolated impurity energy levels above the valence ban irradiation and UV light excites electrons in both the VB and the impurity energy levels, but illumination with visible light only excites electrons in the impurity energy level.

(c) Oxygen vacancies: Gonzalez-Torres et al. [74] concluded that oxygen-deficient sites formed in the grain boundaries are important for the emergence of vis-activity and nitrogen doped into oxygen-deficient sites are important as a blocker for reoxidation.

Codoping: Heterostructuring the $TiO_2$-001 by codoping with two or more dopants is reported to achieve significant synergistic effects compared to their single ion doped or undoped $TiO_2$ counterparts [75,76]. The strong interaction between these dopants within the $TiO_2$ matrix alters the charge carrier transfer-recombination dynamics and shifts the band gap absorption to the visible region [77]. In the case of Ni and N [78], N broadens the absorption profile, improving the photoutilization of $TiO_2$, and generates more electron–hole pairs, while Ni doping restrains the increase of grain growth and leads to crystal expansion, retarding the recombination of charge carriers and thus resulting in the faster degradation of MO compared with single ion doping or undoped $TiO_2$ under UV light.

Noble metal ion doping (surface doping): Noble metal doping creates high Schottky barriers in the $TiO_2$ lattice which acts as electron traps by promoting interfacial electron transfer and thereby suppressing the recombination [79]. Noble metal islands deposited on the $TiO_2$ surface show surface plasmon resonance (SPR) due to which the visible light absorption of the catalysts increases drastically [80]. It has been reported in the literature that the noble metals such as Pt [81], Au [82] and Pd [83] deposited on $TiO_2$-001 have high Schottky barriers among the metals and thus act as electron traps and increase visible light absorption through SPR effect.

### 2.3.2. Coupling

Coupling TiO$_2$-001 with carbon-based materials: Carbon-based materials such as carbon nanotubes, carbon quantum dots, fullerenes, graphene, graphene oxide, reduced-graphene oxide and graphene quantum dots have been proven as effective materials to couple with 001-TiO$_2$. This coupling further suppresses the recombination of the photogenerated electron–hole pairs. These carbon-based materials have the following characteristics, which make them an ideal candidate for coupling with TiO$_2$-001 materials [29].

(a)  High surface area: High specific surface area with many active sites on the surface to boost photocatalytic reactions as compared to their bulk counterpart.
(b)  $\pi$-conjugated structures: This leads to fast electron transfer and promote the separation of electron−hole pairs on the photocatalyst surface.
(c)  Excellent support matrix: Carbon-based materials can form efficient heterojunctiond by having intimate contact between them and TiO$_2$-001 materials, such as point-to-face contact (0 D) for carbon/graphene quantum dots, line-to-face contact for carbon nanotubes (1 D) and face-to-face contact for graphene (2 D) as presented in Figure 7. This is more beneficial for the rapid charge transfer and better catalytic dispersion to enhance the photocatalytic activity [84,85].

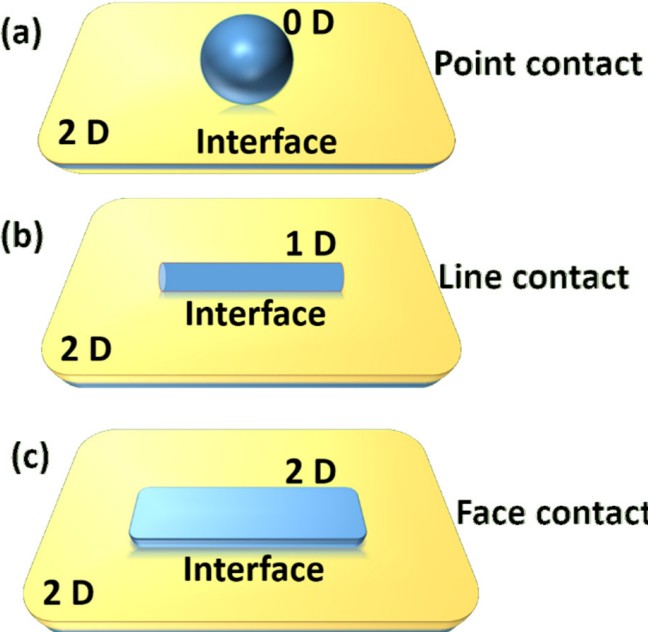

**Figure 7.** Schematic diagram of interfaces of (**a**) 0 D (**b**) 1 D and (**c**) 2 D materials with TiO$_2$-001 nanosheet surfaces [84].

There is a cornucopia of examples that explore carbon-based, material-based nanocomposites which could advance fabrication photocatalysis and Li-ion battery anodes. Some of them are discussed in detail in the application section of this review [31,86–88].

Coupling TiO$_2$-001 with other semiconductors: TiO$_2$-coupled semiconductor photocatalysts are a novel approach to achieve more efficient charge separation [89]. It increases the lifetime of the charge carriers (as shown in Figure 8), and enhances interfacial charge transfer to adsorbed substrates [90]. At the same time, physical and optical properties of the coupled nanocomposites are greatly modified.

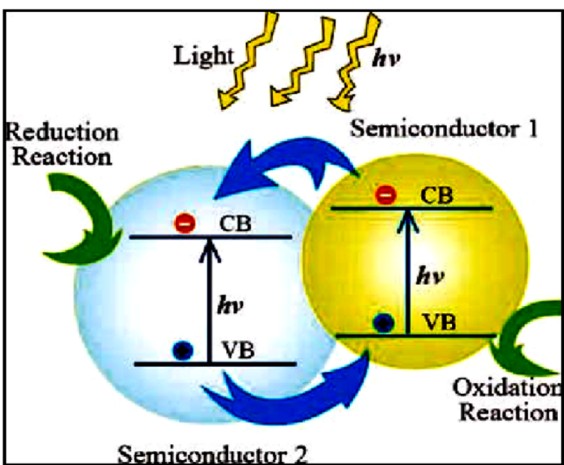

**Figure 8.** A pictorial representation of the coupling of a semiconductor with TiO$_2$.

Metal oxides, such as Cu$_2$O [91], Fe$_3$O$_4$ [92], Cds [93], MoO$_3$ [94], SnO$_2$ [95]], and so on, have been considered for band gap engineering of TiO$_2$ which are discussed in detail in application section.

Among these oxides, low band gap Cu$_2$O is used as a sensitizer to enhance visible light absorption [96], whereas other large band gap oxides (SnO$_2$) are coupled with TiO$_2$ for extrinsic trapping of photogenerated charge carriers to enhance photoactivity. Coupling TiO$_2$ with SnO$_2$ has attracted much attention for Li-ion battery applications, as the theoretical lithium storage capacity is of SnO$_2$ 790 mAhg$^{-1}$, which is more than twice that of graphite (370 mA hg$^{-1}$) [31].

### 2.3.3. Crystal Growth

During synthesis, the formation of faceted TiO$_2$ crystals is always influenced by different environmental growth conditions (either acidic or alkaline conditions). By simply manipulating the pH or concentration of the HF solution used, the size, shape and morphology of {001} faceted TiO$_2$ crystals could be effectively controlled [97,98]. As a result, the morphologies of the as-synthesized anatase vary with surface chemistry and thus the photoactivity [99]. Various morphologies reported for TiO$_2$-001 are given in Figure 9.

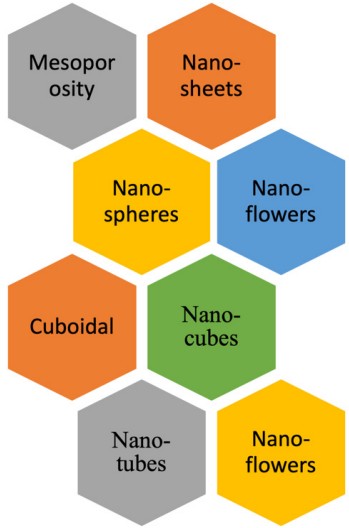

**Figure 9.** Various morphologies reported for TiO$_2$-001 structures.

In this case, efforts are being made to synthesize TiO$_2$-001 with high surface areas and porosity. Two-dimensional structures like nanoplates, nanosheets and hierarchical

structures like nanoflowers provide high surface area and active sites for catalytic adsorption [100]. Porous structures can provide a wide surface area with cavity effect [101,102]. This photon energy entrapment in a porous structure may further expand the photophysical effect in applications, such as photocatalysis and Li-ion batteries [20,103].

### 3. Synthesis of TiO₂-001 and TiO₂-001-Based Composites

According to the Wulff construction, the equilibrium shape of anatase crystals is a slightly truncated bipyramid having eight {101} and two {001} facets (Figure 10). This is the most popular crystal shape occurring in nature with a typical degree of truncation (B/A) in an order of 0.3–0.4 range, giving less than 10% of {001} exposed facets [30].

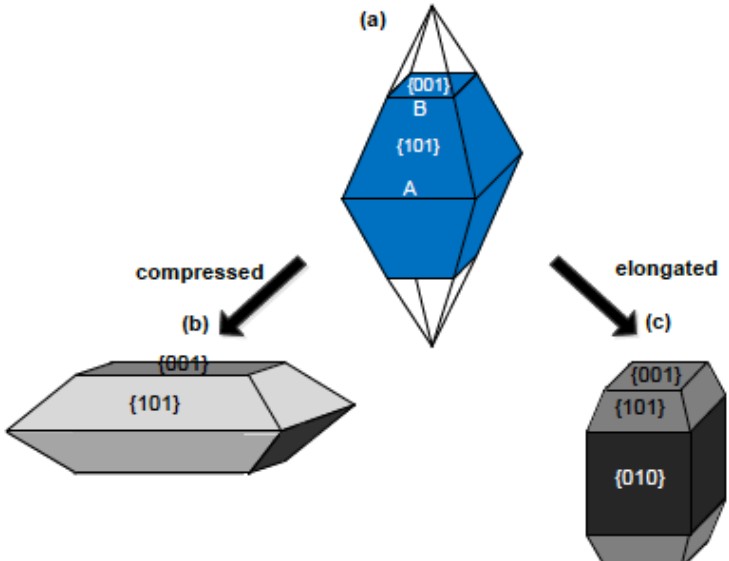

**Figure 10.** (**a**) Shape of anatase TiO₂ under equilibrium conditions, (**b**,**c**) shape of anatase TiO₂ under non-equilibrium conditions. The side lengths labelled as A and B are used to estimate the degree of truncation (B/A) and also calculate the percentage of exposed {001} facets. (Reprinted with the permission of ref. [30], copyright 2014 © John Wiley and Sons).

### 3.1. Synthesis of TiO₂-001 Nanomaterials

To synthesize TiO₂ nanocrystals with dominant 001 facets, the crystal growth should be confined within the kinetically controlled regime under non equilibrium conditions [42]. The key in controlling the percentage of any exposed facets is to alter their relative stability during crystal growth, which could be achieved with the help of surface adsorption of capping agents [37]. In short, a general approach to control the percentage of any crystal facets present is to grow the crystal in the presence of a capping agent (adsorbing species) which can alter its surface energy. Thus, the role of capping agents is crucial to selectively adsorb and reduce the surface free energy of materials with more active sites inhibiting the crystal growth along the corresponding direction [104].

To investigate the effects of various adsorbates (capping agents) on the stability of {001} facets, Yang et al. systematically explored a range of 12 non-metallic adsorbate atoms X (X = H, B, C, N, O, F, Si, P, S, Cl, Br and I) by performing first-principles calculations. The calculated surface energies ($\gamma$) for all the 12 adsorbates are illustrated in Figure 11a [43].

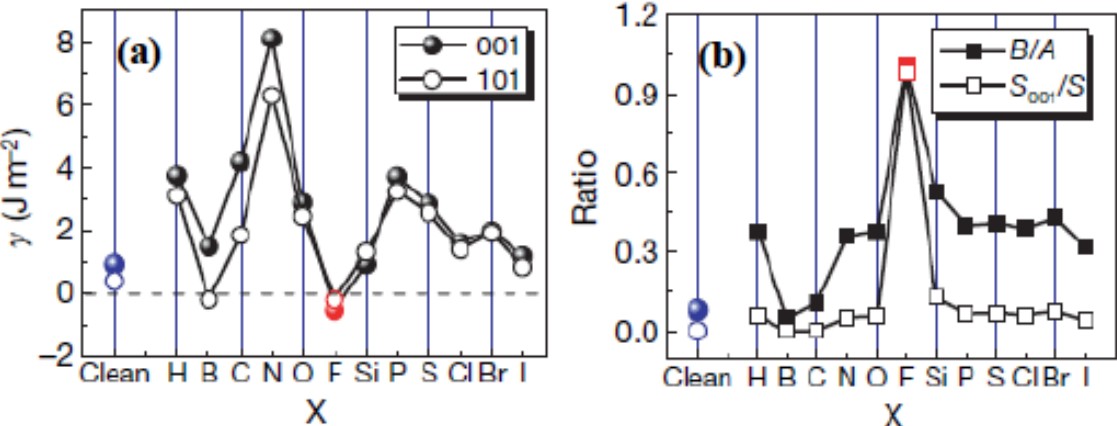

**Figure 11.** (**a**) Calculated energies of the {001} and {101} facets surrounded by X atoms (X = H, B, C, N, O, F, Si, P, S, Cl, Br and I). (**b**) Plots of the optimized value of B/A and percentage of exposed {001} facets for single anatase crystal with various adsorbate atoms X. (Reprinted with permission from ref. [43], Copyright 2008 © Nature Publishing Group).

Referring to Figure 11a, two conclusions were: (1) F-terminated anatase TiO2 surfaces have the lowest $\gamma$ for both {001} and {101} facets among the 12 non-metal-terminated surfaces, and (2) {001} facets are preferential and more stable compared to {101} facets for F-terminated anatase TiO2 crystals. On the basis of these experiments, they verified that there is strong preferential interaction between fluorine and the {001} facets of anatase TiO2 crystals which enhances the relative stability of the {001} facets during the crystal growth. Hence, they used HF as a capping agent and $TiF_4$ as raw material in a hydrothermal reaction to synthesize TiO2-001, which yielded $TiO_2$ with 47% {001} facets. This breakthrough stimulated many research efforts towards the synthesis of TiO2-001 with an increased percentage of {001} facets using HF as a capping agent [43].

Along with the HF, Yang and co-workers used 2-propanol to synthesize high quality anatase TiO2 crystals with 64% of exposed {001} facets. They discovered that alcohols not only acted as a reaction medium, but also as a protecting agent to manipulate the isotropic growth of anatase TiO2-001 nanomaterial [105,106].

Though there are many investigations into $TiO_2$-001 synthesis using HF as a capping agent, the use of HF is extremely corrosive. Hence, many researchers are now adopting HF-free synthesis methods which are comparatively safe and environmentally friendly. Some of them are mentioned in Table 1.

**Table 1.** Fluorine-mediated synthesis routes.

| Precursor | Capping Agent | Synthesis Route | Solvent | % Of 001 Facets | Reference |
|---|---|---|---|---|---|
| $TiF_4$ | HF | Hydrothermal | Water | 47 | [43] |
| Titanium butoxide | HF | Hydrothermal | Water | 89 | [107] |
| $TiF_4$ | HCl | Hydrothermal | Water | >90 | [108] |
| $TiF_4$ | Disodium EDTA | Hydrothermal | Water | 22.8 | [109] |
| $TiCl_4$ | $NH_4F$ and 2-propanol | Solvothermal | Alcohol/water | - | [110] |
| $TiOSO_4$ | $NH_4HF_2$ | Solvothermal | Water | - | [111] |
| $TiCl_4$ | $NaBF_4$ and NaF | Solvothermal | HCl/water | 55% | [112] |
| $(NH_4)_2TiF_6$ | $H_3BO_3$ and 2-propanol | Solvothermal | $H_3BO_3$ and 2-propanol | | [113] |

However, F-containing compounds generate toxic and corrosive substances at elevated temperatures in hydrothermal synthesis [114]. Moreover, due to the strong interaction between TiO2-001 crystal surface and $F^-$ ions, removing them is very difficult. Thus,

developing a fluorine-free synthesis methodology is very necessary. To date, there are some papers reporting fluorine-free strategies for fabricating anatase $TiO_2$-001, which are listed in Table 2.

**Table 2.** Fluorine-free synthesis routes.

| Precursor | Capping Agent | Synthesis Route | Solvent | Reference |
|---|---|---|---|---|
| $TiO_2$ powder | Oleic acid Oleyamine | Solvothermal | Water vapor | [115] |
| Amorphous $TiO_2$ | Polyvinyl pyrrolidone with acetic acid | Electrospun followed by hydrothermal method | 9.6% | [116] |
| $TiCl_4$ | Ethylene glycol | Hydrothermal | 18% | [117] |
| K-titanate wires | $(NH_4)_2(CO_3))$ | Hydrothermal | 60% | [114] |
| Ti-isopropoxide | DETA -and isopropyl alcohol | Solvothermal | ~100% | [58] |
| Tetrabutyl titanate | $H_2SO_4$ | Hydrothermal | ~100% | [118] |
| Ti-isopropoxide | DEA-TBAH | Hydrothermal | - | [119] |
| Ammonium titanate nanowires | HMTA | Hydrothermal | - | [120] |
| $TiCl_3$ | $H_2O_2$ | Hydrothermal | - | [121] |

DETA—diethylene triamine; TBAH—tetrabutyl ammonium hydroxide; HMTA—hexamethylene tetraamine.

Among the methods to grow $TiO_2$-001, the hydrothermal method is more often used, as can be noticed in Tables 1 and 2. Some other methods which can be used are the microwave heating method [13], chemical vapor deposition (CVD) [122] and the electrochemical anodization method [123]. Among these methods, CVD is often used to synthesize high-density $TiO_2$-001thin films with preferred orientation. In a recent work, Lee and Sung reported the synthesis of $TiO_2$ nanosheets on silicon and silicon-coated substrates using the CVD method. They discovered that the silicon vapours can suppress the growth of anatase crystals into [1] orientation, resulting in the formation of 2D-nanosheets with dominant {001} facets [124].

### 3.2. Synthesis of $TiO_2$-001-Based Composites

As explained in the previous section, there are several types of $TiO_2$-001-based composites which mainly include $TiO_2$-001 doped with metal or non-metal, noble metal deposited $TiO_2$-001 nanomaterials and $TiO_2$-001 coupled with other semiconductors or carbon-based materials.

Hydrothermal and solvothermal methods are the most common methods used for the synthesis of doped $TiO_2$ composites. For doping a particular metal/non-metal ion, a soluble precursor containing that ion is added during the hydrothermal reaction. However, fluorine-rich hydrothermal methods yield products with very high crystallinity, making the inclusion of dopant species into the structural framework difficult [125]. Hence, very specific dopant precursors have to be used for the synthesis of doped $TiO_2$-001 composites. For example, for nitrogen doping, titanium nitride has been used as both a Ti precursor and an N doping source [68]. $TiS_2$ and TiC have been used as a sulfur and carbon precursor, respectively [126].

For noble metal deposition, $TiO_2$-001 is synthesized by a hydrothermal method and noble metal is deposited on it by photochemical or chemical reduction process. In this context, core-shell morphology is highly preferred due to their controllable chemical and colloidal stability within the shell and charge transfer between metal cores and semiconductors [104]. Wu and co-workers successfully fabricated Au/$TiO_2$-001 nanomaterials in a core-shell form in which $TiO_2$ nanomaterials have truncated wedge-shaped morphology. A schematic of the synthesis method is shown in Figure 12 [127]. They demonstrated that

the morphological evolution of TiO2 shells is highly dependent on the concentration of $F^-$ ions produced during the reaction. The F- ions play a dual role of producing well-defined wedge-like TiO2 shells and directing growth of truncated crystal {001} facets.

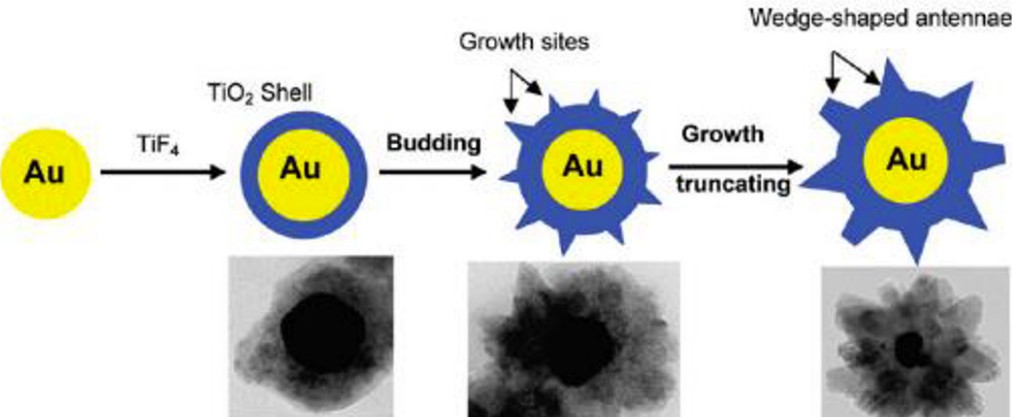

**Figure 12.** Schematic illustration of the proposed formation process of core-shell $Au/TiO_2$ nanoparticles with truncated wedge-shaped morphology. (Reprinted with the permission from ref. [127], copyrights 2009 © American Chemical Society).

$TiO_2$-001 coupled with other semiconductors could also be synthesized by only the hydrothermal method or it can be used in combination with a sol gel and template method or the layer-by-layer (LBL) self-assembly method. Shen et al. developed a magnetic composite consisting of $Fe_3O_4/SiO_2/Au/TiO_2$ sandwiched layers. A schematic of synthesis is shown in Figure 13. Considering $Fe_3O_4$ as a core nanomaterial, a $SiO_2$ layer is deposited on it by sol gel synthesis using ethylene glycol, tetraethyl orthosilicate (TEOS). Then, the Au nanomaterials are coated on this assembly using electrostatic layer-by-layer and interfacial growth methods. Finally, the $TiO_2$ layer is assembled using the hydrothermal method. This catalyst shows higher photocatalytic activity in visible light and magnetic separability. Moreover, these sandwich-structured photocatalysts can be applied to other catalytic systems such as dye decoloration, water decomposition, hydrogen generation and so on [92].

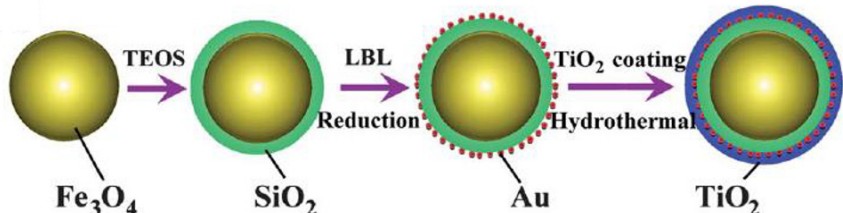

**Figure 13.** Schematic illustration of the synthetic process of the Fe3O4/SiO2/Au/TiO2 photocatalysts. (Reprinted with the permission of ref. [92], copyrights 2012 © Royal Society of Chemistry).

Similarly, for the synthesis of $TiO_2$-001 composites with carbon-based materials, hydrothermal or solvothermal methods have been used extensively. This process is usually followed by microwave heating or ultrasonication for the uniform distribution of $TiO_2$ materials on graphene template. A schematic of the process is shown in Figure 14 [128].

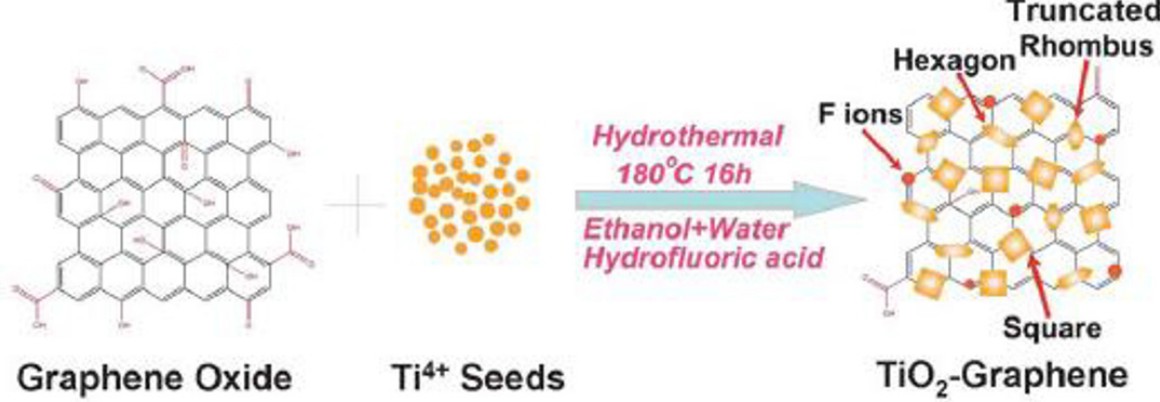

**Figure 14.** Schematics for the preparation of graphene/TiO$_2$ nanocomposites by a hydrothermal treatment in the solvent of ethanol, water and HF. (Reprinted with the permission of ref. [128], copyrights 2012 © Royal Society of Chemistry.

## 4. Applications

### 4.1. TiO$_2$-001-Based Composites for Photocatalytic Applications

Most of the photocatalytic reactions occur on the surface of a photocatalyst and involve the transfer of photogenerated charge carriers. Therefore, the surface electronic structure plays an important role in the photocatalytic activity of TiO$_2$ nanomaterials. Theoretical predictions suggest that the {001} facet has the highest surface energy and special atomic structure. Hence, {001} faceted TiO$_2$ nanoparticles show superior performance in photodegradation, water splitting and reduction of CO$_2$.

Additionally, it is noted that TiO$_2$ nanosheets with {001} facets can reduce the recombination rate of the photogenerated electron hole pairs. To further increase the charge separation efficiency and visible light absorption ability, TiO$_2$-001 nanomaterials are coupled with other materials. Recently, various allotropes of carbon materials, including active carbon, carbon nanotubes (CNTs) graphene, graphene oxide (GO) and reduced graphene oxide (rGO) have been combined with TiO$_2$, opening a door to a research frontier for these new, smart composite materials [87,129]. Among all, graphene is the star material for modifying the properties of the photocatalyst material. Figure 15 displays the major roles of graphene in the resulting composite photocatalysts [29]. It is believed that TiO$_2$-001–graphene composites are high-performance candidates in photocatalytic applications. Some of these composites and their applications are listed in Table 3.

**Table 3.** Photocatalytic applications of TiO$_2$-001-based composites.

| Catalysts | % Of 001 Facets | Synthesis Methods (TiO$_2$ Composites) | Applications | Ref |
|---|---|---|---|---|
| TiO$_2$ on graphene sheets | 61 | Fluorine-mediated microwave–hydrothermal (ethanol–water) | H$_2$ production | [130] |
| TiO$_2$ on graphene sheets | 64 | Fluorine-mediated solvothermal method | Degradation of RhB | [131] |
| Graphene–TiO$_2$ composites | ~100% | Fluorine-mediated solvothermal method | Degradation of methylene blue | [132] |
| Graphene–TiO$_2$ | - | Flurine-mediated solvothermal (methnol–water) | H$_2$ production | [133] |
| Graphene–TiO$_2$ | - | Solvothermal method without any capping agent | Degradation of methylene blue | [28] |

**Table 3.** *Cont.*

| Catalysts | % Of 001 Facets | Synthesis Methods (TiO$_2$ Composites) | Applications | Ref |
|---|---|---|---|---|
| Graphene aerogels–TiO$_2$ | 50% | Glucose-mediated hydrothermal method | Degradation of methyl orange | [134] |
| Graphene foam–TiO$_2$ | - | Fluorine-mediated solvothermal method (ethanol–water) | Degradation of Cr (IV), methyl orange and phenol | [135] |
| RGO–TiO$_2$ | - | Fluorine-mediated solvothermal method | Degradation of basic violet | [136] |
| RGO–TiO$_2$ | - | (DETA)-mediated by solvothermal method | Photo-electrochemical current | [137] |
| Carbon dots–TiO$_2$ | - | Fluorine-mediated hydrothermal method | H$_2$ production | [138] |
| Carbon fibres–TiO$_2$ | 40−92 | Fluorine-mediated hydrothermal method | Degradation of methyl orange | [139] |
| Mesoporous-carbon nitride C$_3$N$_4$-TiO$_2$ | 80.6 | TiO$_2$ fluorine-mediated hydrothermal, composite-mechanical stirring | Degradation of methylene blue | [140] |
| CNTs–TiO$_2$ | 75% | Fluorine-mediated hydrothermal method | Degradation of RhB | [141] |
| Fe$_3$O$_4$ @TiO$_2$ core-shell | - | Combination of template and solvothermal method | Degradation of methylene blue | [142] |
| SrTiO$_3$–TiO$_2$ | - | Fluorine-mediated hydrothermal method | Degradation of RhB dye | [143] |
| Bi$_4$O$_5$Br$_2$–TiO$_2$ | - | TiO$_2$: Fluorine-mediated hydrothermal method Composite: Electrostatic self-assembly | NO removal | [144] |
| Au/TiO$_2$ | 58−77 | TiO$_2$: Fluorine-mediated hydrothermal method Au photodeposition: green method using citrate ions | Photocurrent density | [82] |
| Pt/TiO$_2$ | 75 | TiO$_2$: Fluorine-mediated hydrothermal method Pt deposition: Photochemical reduction | Hydrogen Production | [81] |
| PdO–TiO$_2$ | - | TiO$_2$: Fluorine-mediated hydrothermal method PdO deposition: Wet impregnation | Degradation of methylene blue and photocurrent measurement | [83] |
| Carbon nitrides–TiO$_2$ | - | Fluorine-mediated hydrothermal method | Photoreduction of CO$_2$ to CO | [145] |
| Polymer–TiO$_2$– Graphene | | Sonochemically assisted fluorine mediated solvothermal method | Photoreduction of CO$_2$ to CO | [146] |
| N-doped TiO$_2$–graphene | | Fluorine-mediated solvothermal method | Photoreduction of CO$_2$ to CH$_4$ | [147] |

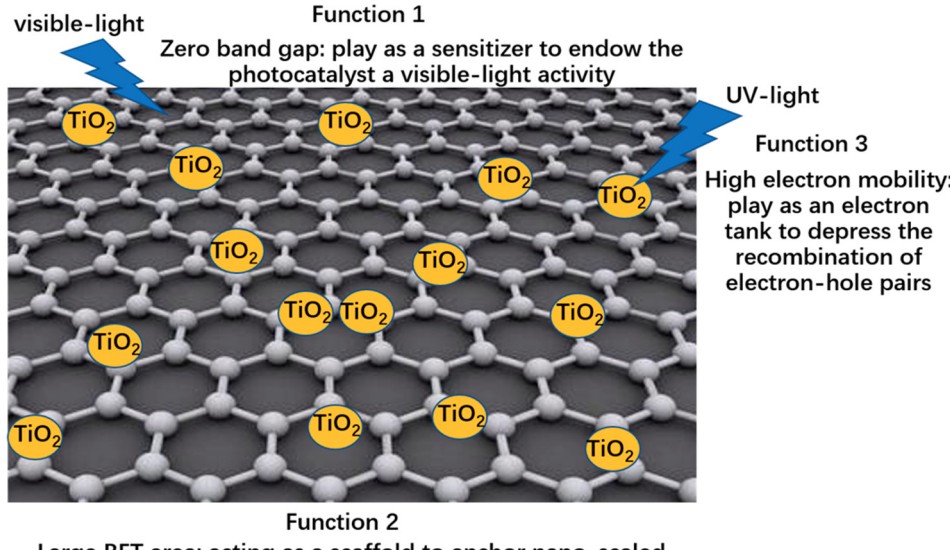

**Figure 15.** Role of graphene in $TiO_2$-001–graphene composites [29].

Xiang et al. deposited hydrothermally synthesized $TiO_2$ nanosheets with exposed {001} facets on graphene sheets by the microwave method. Among all the composites with 0.2 to 5% graphene content, 1% showed almost 41-times higher photoactivity than the pure $TiO_2$. In the composite material, graphene can act as an efficient charge separator due to the ideal heterojunction formed between the graphene and $TiO_2$ nanomaterials. As can be seen from the band diagram of $TiO_2$ and graphene, the valence band of graphene lies below (−0.24 V) the conduction band of anatase $TiO_2$ (Figure 16) [148] which favors the electron transfer from the conduction band (CB) of $TiO_2$ to graphene sheets. Furthermore, the lower potential value of valence band of graphene is 0.08 V, which is slightly higher than the reduction potential of $H^+$ (0 V), which favors the reduction $H^+$ to $H_2$, thus enhancing photocatalytic $H_2$-production activity [130].

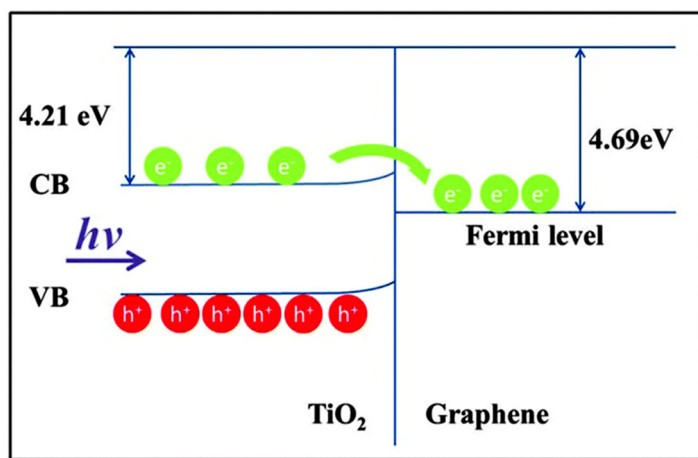

**Figure 16.** Band diagram of $TiO_2$ and graphene heterojunction [148].

Similarly, Sun et al. fabricated nano-sized anatase $TiO_2$-001 nanosheets by the solvothermal route using HF as the morphology controlling agent. These nanosheets were deposited on graphene sheets by the molecular grafting method. These photocatalysts show 2.8-times enhancement in the photoactivity compared to commercial $TiO_2$ photocatalysts, which is attributed to the effective charge separation of graphene and the high catalytic activity of {001} facets [131]. Wang et al. demonstrated that the 001 faceted

$TiO_2$-graphene composites show higher photocatalytic activity than the Degussa $TiO_2$ for methylene blue degradation reaction. The enhancement in the photoactivity can be attributed to the electron transfer via Ti–O–C between $TiO_2$ and C interaction which greatly retards the recombination of photoinduced charge carriers. The optimum amount of graphene is found to be 1 wt %, at which the $TiO_2$/graphene sample displays the highest reactivity [132].

Despite higher photocatalytic efficiency, the biggest problem hindering the applications of $TiO_2$–graphene composites is their stability. Generally, it is observed that the forces of interaction between $TiO_2$ and graphene are weak. If the interaction is weak, the composite formation turns out to be inefficient, hampering the transfer of electrons and Li-ions. To avoid this problem, Qiu et al. developed ultralight 3D-graphene aerogels (GAs) which have high surface area, massive appearance and hydrophobic properties. $TiO_2$ crystals with 001 facets show more stable and higher dispersions on these Gas, which contributes towards the recyclability of these photocatalysts. In this method, glucose is used as a linker and dispersant between $TiO_2$ and GAs. The strong interaction between $TiO_2$ and GAs, the facet characteristics, the high electrical conductivity and the three-dimensional hierarchically porous structure of these composites results in highly efficient charge separation and high surface area, showing enhanced photoactivity. In addition to this, these aerogels can better adsorb organic pollutants and can provide multidimensional electron transport pathways, implying a significant potential application for photocatalysis [134].

On similar grounds, Chen et al. developed free standing, 3D graphene foam through leavening GO films. Then, 001 faceted $TiO_2$ arrays are integrated into these graphene structures through a facile vacuum filtration process combined with a solvothermal method (Figure 17). This results in the uniform distribution of $TiO_2$ nanosheet arrays into interlayers of graphene foam, forming a 3D hierarchical porous structure. These structures show superior photoactivity towards MO dye degradation, which is 8.7-times higher than the commercial catalysts, i.e., P25. Moreover, these composites are very stable, recyclable and can be reused after 12 runs with high efficiency. The unsaturated coordination atoms and dangling bonds that existed on {001} facets of $TiO_2$/graphene foam were highly reactive for adsorbing active oxygen species and facilitated the surface catalytic reaction by providing abundant reactive sites [135]. Hence, these composites are highly appreciated for their higher photoactivity, easy separation and reusability.

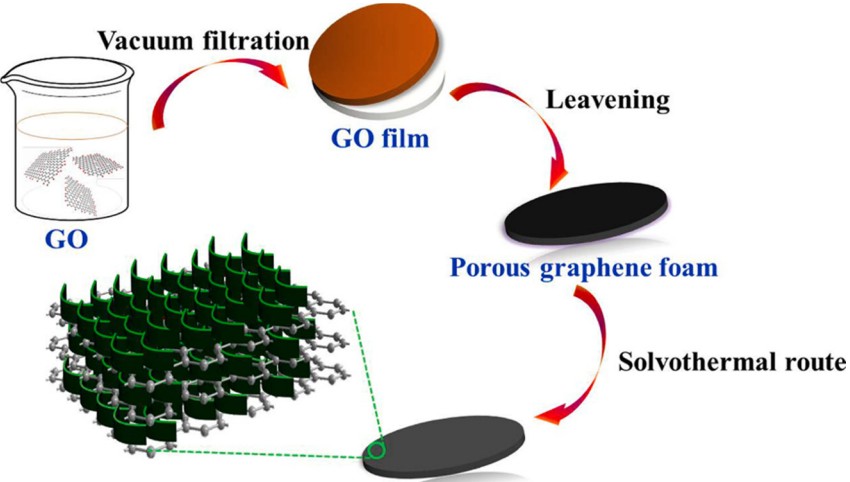

**Figure 17.** Synthesis route for $TiO_2$-001–graphene foam composite. (Reprinted with the permission of ref. [135], copyrights 2021© Elsevier).

Graphene oxide (GO) has a similar hexagonal carbon structure to graphene but it also contains hydroxyl (–OH), alkoxy (C–O–C), carbonyl (C=O), carboxylic acid (–COOH) and

other oxygen-based functional groups. GO can be treated by several methods to synthesize reduced graphene oxide (rGO or RGO) in efforts to minimize the number of oxygen groups and achieve properties closer to those of pristine graphene. Liu et al. incorporated rGO into 001 faceted flower-like $TiO_2$ via solvothermal method. The results indicated that the photocatalytic activities of these composites were significantly improved following the incorporation of RGO, particularly under visible light irradiation [136]. Thien et al. fabricated $TiO_2$ with highly exposed {001} facets decorated on reduced graphene oxide (RGO) sheets by simple solvothermal route. These composites were sintered at 300, 400 and 500 °C. The results indicated that the composites showed highest photocurrent densities at 500 °C, which is attributed to the reduced recombination of electron–hole pairs and improved light harvesting efficiency [137].

Along with graphene and rGO, other carbon-based materials such as carbon dots and carbon nanotubes can also be coupled with the 001 faceted $TiO_2$ for higher photoactivities. Sui et al. fabricated carbon quantum dot–$TiO_2$ composites and Wang et al. prepared carbon nanotubes–$TiO_2$ composites by a fluorine-mediated hydrothermal method [138,141]. Both composites show higher photocatalytic activity than the pure $TiO_2$ nanomaterials. Guo et al. introduced carbon fibers (CF), which are flexible, conductive and stable substrate to grow 001 faceted $TiO_2$ nanosheets [139]. These $TiO_2$-001/CF hybrid structures exhibited 3.38-fold improved photocatalytic degradation of methyl orange (MO) than commercial catalysts and showed excellent stability under ultraviolet–visible (UV–vis) light irradiation. Graphitic carbon nitride (g-$C_3N_4$), with a layered structure, is the new class of materials having properties like graphene. It is a polymeric semiconductor consisting of C, N and some impure H, connected via tris-triazine-based building blocks. Due to the presence of N and H atoms it has a basic tendency of H-bonding. Due to its electron-rich properties, it has a capacity to complement carbon in various photochemical applications [149]. Liu et al. developed a (g-$C_3N_4$- $TiO_2$-001) photocatalyst via a facile ultrasonic dispersion method. All these composites show higher photocatalytic activities which could be attributed to the synergistic effect of $TiO_2$ and C, which greatly retard the recombination of photoinduced electrons and holes. Furthermore, the carbon-based materials serve as a photosensitizer and sensitize $TiO_2$-001 through the newly formed Ti–O–C bond, leading to superior photocatalytic activity [140].

Chen et al. fabricated hollow ellipsoidal structures made up of three-layered structures which act as efficient photocatalysts with magnetic separability. In this complex, hematite (a-$Fe_2O_3$) nanospindles act as the starting templates, on which the silica ($SiO_2$) layer is coated by a hydrothermal process. Finally, ultrathin anatase $TiO_2$ nanosheets (NSs) with exposed (001) facets are deposited on this assembly by the solvothermal process, which acts as a shell. The middle $SiO_2$ layer is then removed by HF treatment, creating hollow ellipsoidal complex structures. The in situ reduction of a-$Fe_2O_3$ to $Fe_3O_4$ imparts magnetic functionality to these photocatalysts, making them magnetically separable under the influence of an applied magnetic field. A schematic of the formation of these three types of structures are shown in Figure 18 [142].

Yue et al. developed $SrTiO_3$/$TiO_2$ heterostructure nanosheets with exposed (0 0 1) facets by in situ hydrothermal reaction. $SrTiO_3$ is a well-known cubic-perovskite-type photocatalyst with a band gap of 3.2 eV and possesses superior chemical stability. As can be seen from Figure 19, the conduction band $SrTiO_3$ lies 200 mV above the conduction band of $TiO_2$. This favours the electron transfer from $SrTiO_3$ to $TiO_2$, leading to the efficient charge separation. Hence, coupling $SrTiO_3$ to $TiO_2$ improves the photocatalytic performances of $TiO_2$ [143].

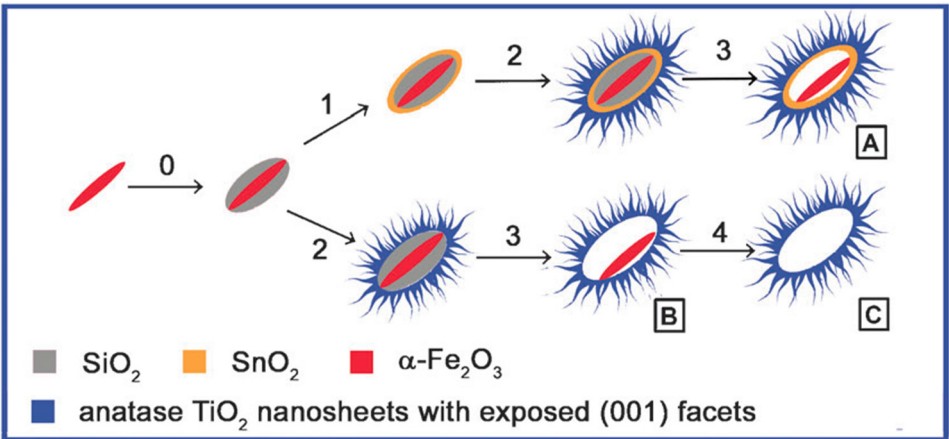

**Figure 18.** Schematic illustration of the fabrication of the three types of complex hollow structures with various procedures: (0) silica coating on a-Fe$_2$O$_3$ spindles; (1) hydrothermal deposition of SnO$_2$ layer; (2) solvothermal formation of anatase TiO$_2$ shell with exposed (001) facets; (3) dissolution of SiO$_2$ using 0.6% HF; (4) removal of a-Fe$_2$O$_3$ spindle using 0.5 M oxalic acid. (Reprinted with the permission of ref. [142], copyrights 2011 © Royal Society of Chemistry).

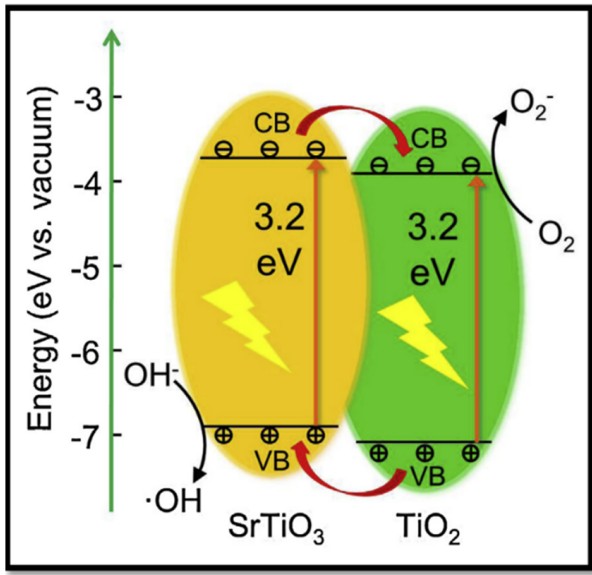

**Figure 19.** Band diagram of TiO$_2$–SrTiO$_3$ heterojunction. (Reprinted with the permission of ref. [143], copyrights 2014 © Elsevier).

Recently, deposition of noble metals such as Au, Ag and Pt on the surface of semiconductors has been recognized as a promising strategy to achieve visible light harvesting through localized surface plasmon resonance (LSPR) photosensization. The electrons in noble metals might be excited by the LSPR with visible light absorption and subsequently transfer to the conduction band of the attached TiO$_2$ semiconductors, increasing the photocatalytic efficiency. Yan et al. fabricated Au-deposited 001 faceted TiO$_2$ using a green photodeposition method. The close contact between the Au and anatase {001} facet leads to enhanced LSPR intensity and promotes the separation of photogenerated electron−hole pairs under visible light irradiation. This increases the photocatalytic activity of the composites in both hydrogen evolution and methylene blue decomposition [82]. Yu et al. developed Pt/TiO$_2$ nanosheets with exposed (001) facets by a hydrothermal route followed by a photochemical reduction deposition of Pt nanoparticles on TiO$_2$ nanosheets. These composites showed higher photoactivities than the commercial photocatalysts which

can be attributed to the plasmonic effect of Pt nanoparticles and synergistic effect of surface fluorination and exposed (001) facets [81]. Lyu et al. explained that the fluorinated 001 surfaces are controversial to the photocatalytic activities. Hence, they removed the surface F atoms bonding with Ti by the hydrogenation method successfully and found that {001} facet dominant $TiO_2$ nanosheets without the terminated F atoms showed dramatic enhancement in the photocatalytic activity. Moreover, the clean (001) surfaces are better for PdO deposition than the fluorine rich surfaces. The PdO attached on {001} facets promoted the separation of charge carriers, and Pd nanoparticles transferred plasmonic-induced electrons to the conduction band of hydrogenated $TiO_2$ under simulated solar irradiation, thus significantly enhancing the photocatalytic activity of Pd-H-$TiO_2$ composites [83].

To address the serious issue of global warming, $TiO_2$-001-based composites have gained considerable attention and much research interest for their advancement of converting $CO_2$ into energy-bearing products without further increasing the concentration of atmospheric $CO_2$. Crake et al. and Wang et al. synthesized carbon nitrides and graphene composites with $TiO_2$-001, which show significant photoactivity for $CO_2$ reduction [145,146].

### 4.2. TiO$_2$-001-Based Composites as a Li-ion Battery Anode Material

Li-ion batteries (LIBs) have aided the electronics revolution by powering portable electronics, mobiles and laptops. They have a potential to transform the transportation sector with electric cars, buses and bikes [150,151]. In addition, Li-ion batteries can also be employed to buffer the intermittent and fluctuating green energy supply from renewable resources, such as solar and wind, to smooth the difference between energy supply and demand [152]. These are one of the most advanced rechargeable batteries, which are lighter in weight, smaller in size and have higher energy density as compared to the traditional rechargeable batteries such as lead acid and Ni-Cd [153]. Li-ion batteries operate by shuttling $Li^+$ and electrons between negative and positive electrode host structures, which are separated by a separator filled with an aprotic electrolyte. In general, the commercial lithium-ion batteries use a graphite–lithium composite, $LixC_6$, as the anode, lithium cobalt oxide, $LiCoO_2$, as the cathode and a lithium-ion conducting electrolyte. When the cell is charged, lithium is extracted from the cathode and inserted at the anode. On discharge, the lithium ions are released by the anode and taken up again by the cathode (Figure 20) [154].

To enhance and optimize the electrochemical properties, intense research has focused on all aspects of these batteries, including improved anodes [155], cathodes [156] and electrolytes [157,158]. A wide range of nanostructured materials have been used as a promising electrode material for high-performance LIBs. $TiO_2$ is a widely studied metal oxide serving as the anode material for LIBs [159–161]. Its most significant advantage is the ability to be charged and discharged at a high current rate (high power). However, $TiO_2$ has a theoretical lithium storage capacity (170 mA h g$^{-1}$) which is lower than that of graphite and very poor electronic conductivity. Hence, proper nanostructure engineering is desired to achieve the highly efficient solid-state diffusion of Li+ ions from $TiO_2$ anodes [162]. Fortunately, the recently discovered anatase $TiO_2$ nanosheets (NSs) with exposed (001) high-energy facets demonstrate high conductivity due to the shorter ion diffusion length (Figure 21) [163]. According to the literature, the energy barrier of $Li^+$ entering the (001) surface of anatase $TiO_2$ is the lowest as compared to the (010) and (101) facets, suggesting much easier migration of Li+ across the (001) surface [164]. Hence, these nanosheets also appear to be effective in improving the lithium storage capacity of $TiO_2$. In addition, 2D $TiO_2$ nanosheets with a high specific surface area can increase the active sites for $Li^+$ insertion, improving its $Li^+$ storage capability.

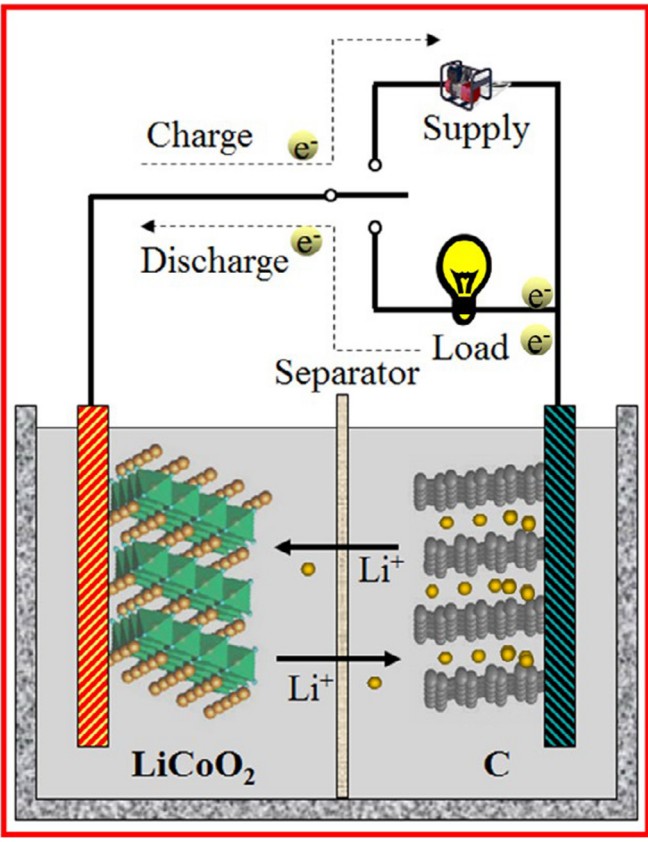

**Figure 20.** Illustration to show the basic components and operation principle of a Li-ion cell [154].

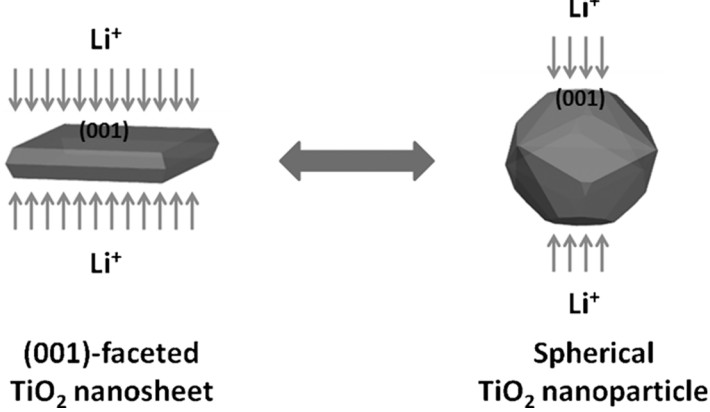

**Figure 21.** Schematic illustration of the facilitation of lithium diffusion in (001) faceted $TiO_2$ nanosheets over spherical $TiO_2$ nanoparticles. (Reprinted with the permission of ref. [163], copyrights 2014 © John Wiley and Sons).

The 2D (001) faceted $TiO_2$ nanosheets are therefore quite promising as anode materials for design of supercapacitor-like LIBs with high energy and power densities. However, the high energy of the (001) facets results in tight aggregation of the $TiO_2$ nanosheets when fabricating the anodes. This reduces the access of $Li^+$ to the active (001) facets and impedes electrolyte penetration inside the structure, deteriorating the electrochemical performances [159]. Hence, $TiO_2$-001 nanomaterials are often coupled with other suitable semiconductors to obtain outstanding conductivity and good structure flexibility. Some of the $TiO_2$-001-based composites are enlisted in Table 4.

**Table 4.** Li-ion battery applications of TiO$_2$-001-based composites.

| Catalysts | Synthesis Method | Capacity at Rate (mAh g$^{-1}$) | Cycling Performance (mAh g$^{-1}$) | Ref |
|---|---|---|---|---|
| Graphene–TiO$_2$-sandwiched structure | Ionic liquid-mediated solvothermal method | 321/1 C | 155 at 10 C | [165] |
| Graphene aerogels–TiO$_2$ | Glucose-mediated hydrothermal method | 605/0.59 C | 50 cycles at 0.59 C | [134] |
| rGO–TiO$_2$ hybrid | Fluorine-mediated hydrothermal method | 250/1 C | 160, 500 cycles at 10 C | [162] |
| TiO$_2$@graphene nanosheets | Wet chemical method | 306 | 400 cycles | [166] |
| rGO@TiO$_2$ nanotubes | Electrostatic interaction and high temp reduction | 263/0.5 C | 500 cycles at 0.5 C | [167] |
| SnO$_2$@TiO$_2$ | Nanotemplate method | 264.8/1 C | 100 cycles at 10 C | [168] |
| MoO$_3$–rutile TiO$_2$ | Fluorine-mediated hydrothermal method | 330/2 C | 100 cycles at 10 C | [169] |
| TiO$_2$–NSs@CNT: | DETA-mediated hydrothermal synthesis | 395/1 C | 120 cycles at 1C | [170] |
| Cu–TiO$_2$ (001) interface | Fluorine-mediated hydrothermal method | 168/1 C | | [171] |
| Carbon supported TiO$_2$ | DETA-mediated solvothermal synthesis | 172.6/1 C | 100 cycles at 1 C | [172] |

1 C = 335 mA g$^{-1}$, diethylenetriamine DETA.

The performance of anodes in Li-ion batteries can be evaluated by a number of parameters, such as specific energy, specific capacity, cyclability and the dis/charging rate. Specific energy (Wh/kg) measures the amount of energy that can be stored and released per unit mass of the battery. It can be obtained by multiplying the specific capacity (Ah/kg) with operating battery voltage (V). Specific capacity measures the amount of charge that can be reversibly stored per unit mass. It is closely related to the number of electrons released from electrochemical reactions and the atomic weight of the host. Cyclability measures the reversibility of the Li-ion insertion and extraction processes, in terms of the number of charge and discharge cycles before the battery loses energy significantly or can no longer sustain function of the device it powers. The rate of charge or discharge measures how fast the battery can be charged and discharged, typically called C-rate. At 1C, the battery is fully discharged, releasing maximum capacity in 1 h.

Among all composites, carbon materials, having high electrical conductivity, large surface area, good structural flexibility and modifiable surface functional groups, have been promising as multifunctional substrates to promote the electrochemical performance of TiO$_2$-001. Liu et al. inserted carbon pillars between TiO$_2$-001 layers. Such a special structure provides open channels for fast Li-ion insertion and extraction. As a result, these composites demonstrate superior Li-ion storage properties and very high reversible capacities at a current rate as high as 50 C [165]. Yu et al. reported the unprecedented lithium storage and electrochemical performance of hierarchically porous TiO$_2$-001/rGO hybrid architecture. They discovered that the tightly packed TiO$_2$ -001 nanosheets prevent the insertion of Li$^+$/Insertion of rGO prevents aggregation and provides many pathways for the electronic Li ion transport. Moreover, (001) faceted TiO$_2$ nanosheets grown in situ on the rGO surface show good mechanical stability, specific surface area and high electron conductivity [162]. Qiu et al. fabricated graphene aerogels (GAs)–TiO$_2$ nanocomposites by a one-step hydrothermal method using glucose as a linker between the two materials (described sect). This TiO$_2$/GAs anode delivers a high capacity of 605 mAhg$^{-1}$ in the first discharge which is much higher than other reported TiO$_2$/carbon electrodes. GAs represent a 3D structured matrix with a large surface area which enhances the dispersion of TiO$_2$ on GAs. Furthermore, their mesoporosity induces shorter diffusion length for Li-ion insertion and extraction, showing magnificent battery performance [134]. Zheng et al. developed TiO$_2$-001 nanotubes wrapped with rGO materials. They deposited a GO layer on TiO$_2$-001

nanotubes by electrostatic interactions and annealed this composite at 700 °C under a 5% H2/Ar mixed atmosphere. During the annealing process, graphene oxide is reduced to graphene and strongly deposited onto the surface of the $TiO_2$ nanotubes. At the same time, the mechanically robust 3D matrix can also effectively buffer the large volume change of the active material. Moreover, $Ti^{3+}$ generated during this process increases the electronic conductivity of the composites. The closely coated rGO layer on the nanotube plays a key role in this preservation of morphology, providing a frame for the tube and preventing the $TiO_2$ from undergoing a phase transformation and breaking into nanoparticles. Because of the advantageous properties of rGO and $Ti^{3+}$, $Li^+$ ions can rapidly diffuse into the electrode via the inner space and the thin walls of the nanotubes. Thus, $rGO/TiO_2$-7ArH exhibits excellent performance as an anode for LIBs [167].

Like graphene, $TiO_2$-001 nanosheets (NSs) can also be grown on CNTs. The role of CNTs in these composite materials is quite well acknowledged, that is, to form a 3D matrix with superior electronic conductivity. At the same time, the mechanically robust 3D matrix can also effectively buffer the large volume change of the active material. The NSs are uniformly grown on the surface of the CNT backbone along its longitudinal axis. Such a unique CNT-$TiO_2$-001 NSs nanocomposite shows good capacity retention and high reversible capacities at high current rates of 5 C and 10 C, proving the positive effect of the CNT backbone [170].

Along with the carbonaceous materials, $TiO_2$-001 coupled with other semiconductors such as $SnO_2$ and $MoO_3$ for better electrochemical performance. Chen et al. fabricated $TiO_2$-001@$SnO_2$ as a core-shell like structure where $SnO_2$ materials act as a template for $TiO_2$ nanomaterial growth (Figure 22B). These ring-like $SnO_2$ materials encapsulate the $SiO_2$ core, which can be seen in Figure 22C. Hence, the $SnO_2$ material acts as a middle layer between the $TiO_2$-001 shell and $SiO_2$ core. Afterward, the $SiO_2$ core is removed by HF treatment, which creates large internal voids inside the $TiO_2$-001@$SnO_2$ composites which can be seen in Figure 22E,F. These hollow structures provide extra space which helps to sustain the volume changes during the Li-ion insertion and removal process. These composites showed excellent performance in a Li-ion battery which can be attributed to the $TiO_2$-001 shell which allows fast transport of Li-ions and large surface area offered by the hollow structures, providing space for Li-ions charge–discharge cycles [168].

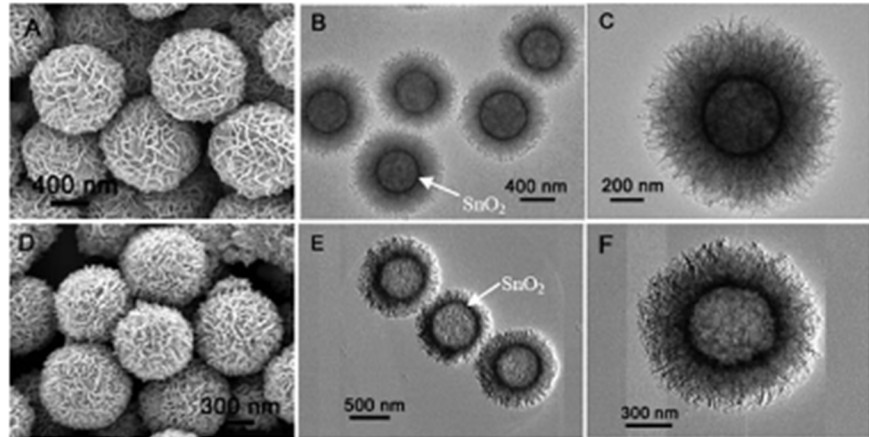

**Figure 22.** (**A**) FESEM image and (**B,C**) TEM images of $TiO_2$-001 shells grown on $SnO_2$@$SiO_2$ core-shell particles. (**D**) FESEM image and (**E,F**) TEM images of $TiO_2$-001 @SnO2 double-shelled hollow spheres. (Reprinted with the permission of ref. [168], copyrights 2010 © Royal Society of Chemistry).

Besides anatase, Chen et al. developed rutile $TiO_2$-001 nanosheets by a unique hydrothermal method, which forms hierarchical microspheres that closely resemble the natural "desert rose" (Figure 23). These nanocomposites are further coupled with the $MoO_3$ to form $TiO_2$–$MoO_3$ hybrid structures. These hierarchical structures provide high surface area for the uniform dispersion of $MoO_3$ on the high-energy (001) facets of rutile

TiO$_2$. In addition, MoO$_3$ nanomaterials provide sufficient stability to preserve the large number of (001) facets, which significantly improves the Li-ion storage performance of the device [169].

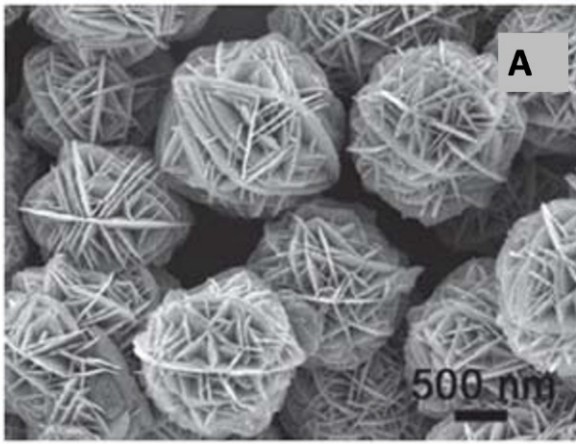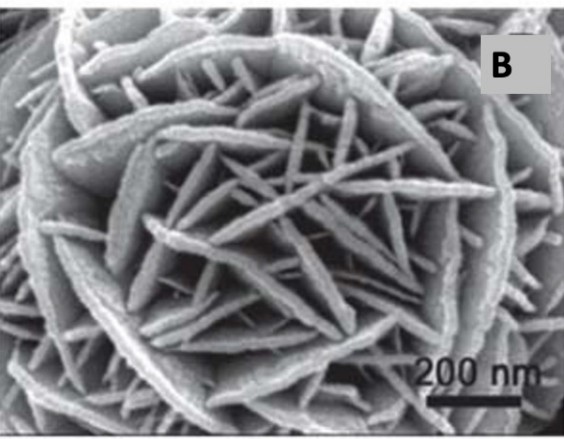

**Figure 23.** (**A**,**B**) Field-emission scanning electron microscopy (FESEM) images of rutile TiO$_2$-001 nanosheets (desert rose morphology). (Reprinted with the permission of ref. [169] copyrights 2011 © Royal Society of Chemistry).

Liu et al. contacted the single TiO$_2$ particle with (010) and (001) facets with the two tungsten probes in a scanning electron microscope and found out that the conduction along the [1] direction is an order of magnitude higher than that along the [10] direction. Based on this finding, a TiO$_2$ anode supported on a copper foil, with its TiO$_2$ (001)–copper interface similar to the TiO$_2$ (001)–tungsten interface, was constructed. This shows a remarkably higher lithium storage capability than a reference TiO$_2$ anode with its interface mimicking the TiO$_2$ (010)–tungsten interface [171].

## 5. Conclusions and Future Scope

Intensive research efforts on TiO$_2$-001-based materials, for their application in environmental remediation and energy conversion, have taken place over the last few years. The continuous advancement in the synthesis methods and modification techniques of TiO$_2$-001 materials have expanded the capabilities of TiO$_2$ to include visible light absorption, suppressed rate of recombination, higher electronic conductivity and higher surface area with more active sites. This expands the possibilities of TiO$_2$ applications into new and diverse areas. Despite the huge development in theoretical studies and experimental investigations involving TiO$_2$-001 and their composites, practical applications in this field are still in the preliminary stages and many challenges exist in several aspects. Some of the challenges are listed below.

1. Developing novel, large-scale synthesis methods

Despite the great success that has been obtained in the controllable synthesis of TiO$_2$-001, there is still room for improvement in terms of quality of the products. Most of the synthesis methods employ fluorine-containing solutions as capping agents, which are highly toxic and corrosive. Hence, fluorine-free, safe and environmentally friendly strategies should be developed to obtain clean 001 facets. Moreover, the large-scale and high-yield production methods must be developed, which is an important prerequisite for practical applications.

2. Coupling TiO$_2$-001 with energy-harvesting materials

The photocatalytic properties of TiO$_2$-001 materials can be amended by the interaction with new materials, which include light harvesters, charge transport materials, additives and interfacial modifiers. Moreover, many efforts must be made to develop large-scale

preparation techniques for high-quality modified $TiO_2$-001 materials i.e., $TiO_2$-001-based composite materials.

3.  Exploring new characterization techniques

Scanning electron microscopy (SEM) and transmission electron microscopy (TEM), as well as small-angle X-ray/neutron scattering (SANS/SAXS), are the widely used techniques to investigate the morphologies of {001} and {101} faceted anatase $TiO_2$ nanocrystals. However, these techniques can only provide selected local morphology information. Moreover, these techniques failed to probe the evolution of $TiO_2$ surface/interface structures in working conditions, which is crucial to study the complex phase transformation and device stability. Hence, more powerful techniques such as near edge X-ray absorption fine structure (NEXAFS) spectroscopy and neutron pair distribution function (PDF) should be used to obtain accurate average morphology and atomistic structures of {001} and {101} faceted anatase $TiO_2$ nanocrystals. Moreover, in situ characterization techniques must be developed to reveal the photocatalysis reaction process.

4.  Theoretical study for understanding photocatalysis mechanisms

The knowledge about the relationship between the dominant 001 facets and their photocatalytic performance and the reaction mechanism needs to be deepened, especially at the molecular level.

Overall, we believe that future research efforts and developments can benefit $TiO_2$-001 composites as a next generation material for energy conversion and environmental remediation.

**Author Contributions:** Conceptualization, A.B. and F.E.; methodology, A.B.; software, F.E.; writing—original draft preparation, A.B.; writing—review and editing, F.E.; visualization, A.B.; supervision, F.E.; project administration, F.E.; funding acquisition, F.E. Both authors have read and agreed to the published version of the manuscript.

**Funding:** This research received no external funding.

**Institutional Review Board Statement:** Not applicable.

**Informed Consent Statement:** Not applicable.

**Data Availability Statement:** Not applicable.

**Conflicts of Interest:** The authors declare no conflict of interest.

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
