# Peer review of "Photocatalysis and Li-Ion Battery Applications of {001} Faceted Anatase TiO2-Based Composites"

_2571-8800, doi:10.3390/j4030038_

Round 1

Reviewer 1 Report

Review of the manuscript J-1343204- to Authors: This is the review type of article dealing with specific anatase for certain applications. On titania one can find about a thousand review articles per year; so my opinion is the review articles have to really did deep to find an angle that may be interesting for readers. The approaches to find and angle can repose on specific time frame (such focus is not really present, so maybe be more specific in explaining which time frame you refer to), specific functionality (not particularly specific but photocatalysis and Li-ion batteries is specific enough), specific modification (grouping of all modifications or subtopics is well done), specific syntheses or specific characterizations or specific monitoring (acceptable enough), or specific insight in the functionality mechanisms (facets are nicely brought up and I think this will be interesting for the readers). Thereof I see some added value in what it seems to be strategically selected a titania photocatalytic and battery describing paper. There are some nice examples drown out of the literature. The authors did a fairly good job in organizing facet focused anatase study.

Some suggestions:

Title: change to - {001} facets dominated anatase based composites: syntheses, properties and application for photocatalysis and Li-ion batteries

Abstract: To begin, the existing strategies for the synthesis of {001} dominated Anatase TiO2 and their composites are discussed. These synthesis strategies include both fluorine mediated and fluorine-free synthesis routes.  – ok

Then, a brief account of the 19 effect of {001} facets on the physicochemical properties of TiO2 and their composites are reviewed, with a particular focus on photocatalysis and Li-ion batteries applications. – why brief, that should be the core of the paper.

Finally, an outlook is given on future strategies discussing the remaining challenges for the development of {001} dominated TiO2 nanomaterial and their potential applications. - ok

Introduction: Offers a good intro.

Discussion: synthetic and characterization parts offer reasonable amount of information, and the same can be said for outlook. However, a too brief amount of information on the effect of facets on the physicochemical properties of TiO2 for photocatalysis and Li-ion batteries applications was given. Since that should be the core of the paper the amount is not adequate. Please, improve, also give a more detailed time frame to covered literature survey. Also decide if you would like to go with titania, amorphous titania, anatase, mixed phases, etc. I think the narrower the better, that’s why I suggested only anatase in the title; remove excess. Minor adjustments and improvements should be done for some of the graphical representations and schematics (Fig 4,6,7).

The English language and style are ok. Overall this is a good job, obviously the authors deal with this research area, I ask for a minor revision. But please don’t take the minor revision easily, and upgrade the paper as suggested.

Reviewer 2 Report

This review details the (001) facets anatase TiO2 materials. This paper contains a lot of information about the synthesis, properties and applications of these materials. The illustrations are good. It fits the scope of J. Nevertheless, this paper contains numerous typos mistakes which need to be corrected before publication. So I suggest major revision before publication. Here is a non-exhaustive list of the typos mistakes:

  • In the Figure captions when you cite the reference, you need to be coherent when you cite the works. Sometimes it is written Ref., Ref, ref. Put the same notation in each caption.
  • When you cite the figures in the text, you need also to be coherent. Sometimes it is Fig or Fig. or Figure or fig… Put the same notation everywhere.
  • The subscript and upperscript need to be checked in all text, escecially for TiO2 (Line 181, 584, 709) or Li+ (line 566, 571, 573, 574).
  • Alone bracket Line 122, 471 or double brackets Line 250, 451.
  • Sometimes there are some problems with the reference numbers, for example, line 233, 302, 469, 539, 616 (brackets missing I guess).
  • “et al.” must be in italic (line 130, 141, 356, 470, 607, 641) and also “i. e.” (443, 697).
  • Line 443 it is P25 not p-25.
  • Line 112, what is no. ? number ?
  • Line 152, number of figure is missing.
  • There are also many problems with the “points”, as example, Line 182 a point is missing, line 220 why a point before “such as”?, Line 231 two points, Line 257 the reference must be before the point.
  • Sometimes, there are points before and after a reference (line 494).
  • Too much space line 390, 393, 399, 406, 407.
  • Line jump Line 423.
  • Line 242 why SnO2 in the middle of the sentence?
  • Line 355 “sol-gel” not “sol gel”.

Round 2

Reviewer 2 Report

Dear Authors,

The comments were well-addressed, the review can be published.

Best regards